# Potential Rapid Diagnostics, Vaccine and Therapeutics for 2019 Novel Coronavirus (2019-nCoV): A Systematic Review

**DOI:** 10.3390/jcm9030623

**Published:** 2020-02-26

**Authors:** Junxiong Pang, Min Xian Wang, Ian Yi Han Ang, Sharon Hui Xuan Tan, Ruth Frances Lewis, Jacinta I-Pei Chen, Ramona A Gutierrez, Sylvia Xiao Wei Gwee, Pearleen Ee Yong Chua, Qian Yang, Xian Yi Ng, Rowena K.S. Yap, Hao Yi Tan, Yik Ying Teo, Chorh Chuan Tan, Alex R. Cook, Jason Chin-Huat Yap, Li Yang Hsu

**Affiliations:** 1Saw Swee Hock School of Public Health, National University of Singapore and National University Health System, Singapore 117549, Singapore; ephwmx@nus.edu.sg (M.X.W.); yha2103@columbia.edu (I.Y.H.A.); tan.sharon@nus.edu.sg (S.H.X.T.); ephrfl@nus.edu.sg (R.F.L.); ephcij@nus.edu.sg (J.I.-P.C.); ephsgxw@nus.edu.sg (S.X.W.G.); ephceyp@nus.edu.sg (P.E.Y.C.); yang_qian@nus.edu.sg (Q.Y.); ephngx@nus.edu.sg (X.Y.N.); ephryks@nus.edu.sg (R.K.S.Y.); haoyi.tan.official@gmail.com (H.Y.T.); ephtyy@nus.edu.sg (Y.Y.T.); ephcar@nus.edu.sg (A.R.C.); jasonyap@nus.edu.sg (J.C.-H.Y.); mdchly@nus.edu.sg (L.Y.H.); 2Centre for Infectious Disease Epidemiology and Research, National University of Singapore, Singapore 117549, Singapore; 3National Centre for Infectious Diseases, Singapore 308442, Singapore; ramona_a_gutierrez@ncid.sg; 4Ministry of Health, Singapore 169854, Singapore; chorhchuan.tan@moht.com.sg

**Keywords:** novel coronavirus, diagnostics, vaccine, treatments, global health, outbreak, MERS-CoV, SARS-CoV

## Abstract

Rapid diagnostics, vaccines and therapeutics are important interventions for the management of the 2019 novel coronavirus (2019-nCoV) outbreak. It is timely to systematically review the potential of these interventions, including those for Middle East respiratory syndrome-Coronavirus (MERS-CoV) and severe acute respiratory syndrome (SARS)-CoV, to guide policymakers globally on their prioritization of resources for research and development. A systematic search was carried out in three major electronic databases (PubMed, Embase and Cochrane Library) to identify published studies in accordance with the Preferred Reporting Items for Systematic Reviews and Meta-Analyses (PRISMA) guidelines. Supplementary strategies through Google Search and personal communications were used. A total of 27 studies fulfilled the criteria for review. Several laboratory protocols for confirmation of suspected 2019-nCoV cases using real-time reverse transcription polymerase chain reaction (RT-PCR) have been published. A commercial RT-PCR kit developed by the Beijing Genomic Institute is currently widely used in China and likely in Asia. However, serological assays as well as point-of-care testing kits have not been developed but are likely in the near future. Several vaccine candidates are in the pipeline. The likely earliest Phase 1 vaccine trial is a synthetic DNA-based candidate. A number of novel compounds as well as therapeutics licensed for other conditions appear to have in vitro efficacy against the 2019-nCoV. Some are being tested in clinical trials against MERS-CoV and SARS-CoV, while others have been listed for clinical trials against 2019-nCoV. However, there are currently no effective specific antivirals or drug combinations supported by high-level evidence.

## 1. Introduction

Since mid-December 2019 and as of early February 2020, the 2019 novel coronavirus (2019-nCoV) originating from Wuhan (Hubei Province, China) has infected over 25,000 laboratory-confirmed cases across 28 countries with about 500 deaths (a case-fatality rate of about 2%). More than 90% of the cases and deaths were in China [1]. Based on the initial reported surge of cases in Wuhan, the majority were males with a median age of 55 years and linked to the Huanan Seafood Wholesale Market [2]. Most of the reported cases had similar symptoms at the onset of illness such as fever, cough, and myalgia or fatigue. Most cases developed pneumonia and some severe and even fatal respiratory diseases such as acute respiratory distress syndrome [3].

The 2019 novel coronavirus (2019-nCoV), a betacoronavirus, forms a clade within the subgenus sarbecovirus of the Orthocoronavirinae subfamily [4]. The severe acute respiratory syndrome coronavirus (SARS-CoV) and Middle East respiratory syndrome coronavirus (MERS-CoV) are also betacoronaviruses that are zoonotic in origin and have been linked to potential fatal illness during the outbreaks in 2003 and 2012, respectively [5,6]. Based on current evidence, pathogenicity for 2019-nCoV is about 3%, which is significantly lower than SARS-CoV (10%) and MERS-CoV (40%) [7]. However, 2019-nCoV has potentially higher transmissibility (R_0_: 1.4–5.5) than both SARS-CoV (R_0_: 2–5) and MERS-CoV (R_0_: <1) [7].

With the possible expansion of 2019-nCoV globally [8] and the declaration of the 2019-nCoV outbreak as a Public Health Emergency of International Concern by the World Health Organization, there is an urgent need for rapid diagnostics, vaccines and therapeutics to detect, prevent and contain 2019-nCoV promptly. There is however currently a lack of understanding of what is available in the early phase of 2019-nCoV outbreak. The systematic review describes and assesses the potential rapid diagnostics, vaccines and therapeutics for 2019-nCoV, based in part on the developments for MERS-CoV and SARS-CoV.

## 2. Material and Methods

### 2.1. Study Identification and Selection

A systematic search was carried out in three major electronic databases (PubMed, Embase and Cochrane Library) to identify published studies examining the diagnosis, therapeutic drugs and vaccines for Severe Acute Respiratory Syndrome (SARS), Middle East Respiratory Syndrome (MERS) and the 2019 novel coronavirus (2019-nCoV), in accordance with the Preferred Reporting Items for Systematic Reviews and Meta-Analyses (PRISMA) guidelines.

There were two independent reviewers each focusing on SARS, MERS, and 2019-nCoV, respectively. A third independent reviewer was engaged to resolve any conflicting article of interest. We used the key words “SARS”, “coronavirus”, “MERS”, “2019 Novel coronavirus”, “Wuhan virus” to identify the diseases in the search strategy. The systematic searches for diagnosis, therapeutic drugs and vaccines were carried out independently and the key words “drug”, “therapy”, “vaccine”, “diagnosis”, “point of care testing” and “rapid diagnostic test” were used in conjunction with the disease key words for the respective searches.

Examples of search strings can be found in Appendix A. We searched for randomized controlled trials (RCTs) and validation trials (for diagnostics test) published in English, that measured (a) the sensitivity and/or specificity of a rapid diagnostic test or a point-of-care testing kit, (b) the impact of drug therapy or (c) vaccine efficacy against either of these diseases with no date restriction applied. For the 2019-nCoV, we searched for all in vitro, animal, or human studies published in English between 1 December 2019 and 6 February 2020, on the same outcomes of interest. In addition, we reviewed the references of retrieved articles in order to identify additional studies or reports not retrieved by the initial searches. Studies that examined the mechanisms of diagnostic tests, drug therapy or vaccine efficacy against SARS, MERS and 2019-nCoV were excluded.

### 2.2. Supplementary Strategies

Given the rapidly developing and commercial nature of information regarding these developments, Google search results from the first five pages of each search term were scanned for relevance. Relevant articles were further identified through snowballing, including a search of press releases and grey literature on vaccine manufacturers’ official websites (Appendix A). A list of potential commercial kits was also provided on 29 January 2020 by Dr. Kim J. Png (personal communication). This list was compiled by Dr. Png from web searches and review of latest business news. The list served to verify and supplement our team’s own search above for review. Clinical trials was also searched via https://clinicaltrials.gov/ to supplement the review of the diagnostics, vaccine and therapeutics of 2019-nCoV, SARS-CoV and MERS-CoV.

## 3. Results

### 3.1. Search Results

An initial search identified a total of 1,065 articles from PubMed, Embase and Cochrane Library. There were 236, 236 and 593 articles related to diagnostics, therapeutics and vaccines, respectively (Figure 1). After reviewing for inclusion and exclusion and the removal of duplications, a total of 27 studies were used for the full review (Figure 1).

A Google search for 2019-nCoV diagnostics (as of 6 February 2020; Appendix A) yielded five webpage links from government and international bodies with official information and guidelines (WHO, Europe CDC, US CDC, US FDA), three webpage links on diagnostic protocols and scientific commentaries, and five webpage links on market news and press releases. Six protocols for diagnostics using reverse transcriptase polymerase chain reaction (RT-PCR) from six countries were published on WHO’s website [9]. Google search for 2019-nCoV vaccines yielded 19 relevant articles.

### 3.2. Diagnostics

With the emergence of 2019-nCoV, real time RT-PCR remains the primary means for diagnosing the new virus strain among the many diagnostic platforms available ([10,11,12,13,14,15,16,17,18,19]; Appendix A). Among the 16 diagnostics studies selected, one study discussed the use of RT-PCR in diagnosing patients with 2019-nCoV [11] (Table 1). The period and type of specimen collected for RT-PCR play an important role in the diagnosis of 2019-nCoV. It was found that the respiratory specimens were positive for the virus while serum was negative in the early period. It has also suggested that in the early days of illness, patients have high levels of virus despite the mild symptoms.

Apart from the commonly used RT-PCR in diagnosing MERS-CoV, four studies identified various diagnostic methods such as reverse transcription loop-mediated isothermal amplification (RT-LAMP), RT-insulated isothermal PCR (RT-iiPCR) and a one-step rRT-PCR assay based on specific TaqMan probes. RT-LAMP has similar sensitivity as real time RT-PCR. It is also highly specific and is used to detect MERS-CoV. It is comparable to the usual diagnostic tests and is rapid, simple and convenient. Likewise, RT-iiPCR and a one-step rRT-PCR assay have also shown similar sensitivity and high specificity for MER-CoV. Lastly, one study focused on the validation of the six commercial real RT-PCR kits, with high accuracy. Although real time RT-PCR is a primary method for diagnosing MERS-CoV, high levels of PCR inhibition may hinder PCR sensitivity (Table 1).

There are eleven studies that focus on SARS-CoV diagnostic testing (Table 1). These papers described diagnostic methods to detect the virus with the majority of them using molecular testing for diagnosis. Comparison between the molecular test (i.e RT-PCR) and serological test (i.e., ELISA) showed that the molecular test has better sensitivity and specificity. Hence, enhancements to the current molecular test were conducted to improve the diagnosis. Studies looked at using nested PCR to include a pre-amplification step or incorporating N gene as an additional sensitive molecular marker to improve on the sensitivity (Table 1). 

In addition, there are seven potential rapid diagnostic kits (as of 24 January 2020; Table 2) available on the market for 2019-nCoV. Six of these are only for research purposes. Only one kit from Beijing Genome Institute (BGI) is approved for use in the clinical setting for rapid diagnosis. Most of the kits are for RT-PCR. There were two kits (BGI, China and Veredus, Singapore) with the capability to detect multiple pathogens using sequencing and microarray technologies, respectively. 

### 3.3. Potential Vaccines

With the emergence of 2019-nCoV, there are about 15 potential vaccine candidates in the pipeline globally (Table 3), in which a wide range of technology (such as messenger RNA, DNA-based, nanoparticle, synthetic and modified virus-like particle) was applied. It will likely take about a year for most candidates to start phase 1 clinical trials except for those funded by Coalition for Epidemic Preparedness Innovations (CEPI). However, the kit developed by the BGI have passed emergency approval procedure of the National Medical Products Administration, and are currently used in clinical and surveillance centers of China [40].

Of the total of 570 unique studies on 2019-nCoV, SARS CoV or MERS-CoV vaccines screened, only four were eventually included in the review. Most studies on SARS and MERS vaccines were excluded as they were performed in cell or animal models (Figure 1). The four studies included in this review were Phase I clinical trials on SARS or MERS vaccines (Table 4) [44,45,46,47]. There were no studies of any population type (cell, animal, human) on the 2019-nCoV at the point of screening. The published clinical trials were mostly done in United States except for one on the SARS vaccine done in China [44]. All vaccine candidates for SARS and MERS were reported to be safe, well-tolerated and able to trigger the relevant and appropriate immune responses in the participants. In addition, we highlight six ongoing Phase I clinical trials identified in the ClinicalTrials.gov register ([48,49]); Appendix A) [50,51,52]. These trials are all testing the safety and immunogenicity of their respective MERS-CoV vaccine candidates but were excluded as there are no results published yet. The trials are projected to complete in December 2020 (two studies in Russia [50,51]) and December 2021 (in Germany [52]).

### 3.4. Potential Therapeutics

Existing literature search did not return any results on completed 2019-nCoV trials at the time of writing. Among 23 trials found from the systematic review (Table 5), there are nine clinical trials registered under the clinical trials registry (ClinicalTrials.gov) for 2019-nCoV therapeutics [53,54,55,56,57,58,59,60,61]. Of which five studies on hydroxychloroquine, lopinavir plus ritonavir and arbidol, mesenchymal stem cells, traditional Chinese medicine and glucocorticoid therapy usage have commenced recruitment. The remaining four studies encompass investigation of antivirals, interferon atomization, darunavir and cobicistat, arbidol, and remdesivir usage for 2019-nCoV patients (Table 5).

Besides the six completed randomized controlled trials (RCT) selected from the systematic review (Table 6), there is only one ongoing randomized controlled trial targeted at SARS therapeutics [92]. The studies found from ClinicalTrials.gov have not been updated since 2013. While many prospective and retrospective cohort studies conducted during the epidemic centered on usage of ribavirin with lopinavir/ritonavir or ribavirin only, there has yet to be well-designed clinical trials investigating their usage. Three completed randomized controlled trials were conducted during the SARS epidemic–3 in China, 1 in Taiwan and 2 in Hong Kong [93,94,95,96,97]. The studies respectively investigated antibiotic usage involving 190 participants, combination of western and Chinese treatment vs. Chinese treatment in 123 participants, integrative Chinese and Western treatment in 49 patients, usage of a specific Chinese medicine in four participants and early use of corticosteroid in 16 participants. Another notable study was an open non-randomized study investigating ribavirin/lopinavir/ritonavir usage in 152 participants [98]. One randomized controlled trial investigating integrative western and Chinese treatment during the SARS epidemic was excluded as it was a Chinese article [94].

There is only one ongoing randomized controlled trial targeted at MERS therapeutics [99]. It investigates the usage of Lopinavir/Ritonavir and Interferon Beta 1B. Likewise, many prospective and retrospective cohort studies conducted during the epidemic centered on usage of ribavirin with lopinavir/ritonavir/ribavirin, interferon, and convalescent plasma usage. To date, only one trial has been completed. One phase 1 clinical trial investigating the safety and tolerability of a fully human polyclonal IgG immunoglobulin (SAB-301) was found in available literature [46]. The trial conducted in the United States in 2017 demonstrated SAB-301 to be safe and well-tolerated at single doses. Another trial on MERS therapeutics was found on ClinicalTrials.gov—a phase 2/3 trial in the United States evaluating the safety, tolerability, pharmacokinetics (PK), and immunogenicity on co-administered MERS-CoV antibodies REGN3048 & REGN3051 [100].

## 4. Discussion

### 4.1. Rapid Diagnostics

Rapid diagnostics plays an important role in disease and outbreak management. The fast and accurate diagnosis of a specific viral infection enables prompt and accurate public health surveillance, prevention and control measures. Local transmission and clusters can be prevented or delayed by isolation of laboratory-confirmed cases and their close contacts quarantined and monitored at home. Rapid diagnostic also facilitates other specific public health interventions such as closure of high-risk facilities and areas associated with the confirmed cases for prompt infection control and environmental decontamination [11,101].

Laboratory diagnosis can be performed by: (a) detecting the genetic material of the virus, (b) detecting the antibodies that neutralize the viral particles of interest, (c) detecting the viral epitopes of interest with antibodies (serological testing), or (d) culture and isolation of viable virus particles. The key limitations of genetic material detection are the lack of knowledge of the presence of viable virus, the potential cross-reactivity with non-specific genetic regions and the short timeframe for accurate detection during the acute infection phase. The key limitations of serological testing is the need to collect paired serum samples (in the acute and convalescent phases) from cases under investigation for confirmation to eliminate potential cross-reactivity from non-specific antibodies from past exposure and/or infection by other coronaviruses. The limitation of virus culture and isolation is the long duration and the highly specialized skills required of the technicians to process the samples.

Where the biological samples are taken from also play a role in the sensitivity of these tests. For SARS-CoV and MERS-CoV, specimens collected from the lower respiratory tract such as sputum and tracheal aspirates have higher and more prolonged levels of viral RNA because of the tropism of the virus. MERS-CoV viral loads are also higher for severe cases and have longer viral shedding compared to mild cases. Although upper respiratory tract specimens such as nasopharyngeal or oropharyngeal swabs can be used, they have potentially lower viral loads and may have higher risk of false-negatives among the mild MERS and SARS cases [102,103], and likely among the 2019-nCoV cases.

#### 4.1.1. Detection of Genetic Material

The existing practices in detecting genetic material of coronaviruses such as SARS-CoV and MERS-CoV include (a) reverse transcription-polymerase chain reaction (RT-PCR), (b) real-time RT-PCR (rRT-PCR), (c) reverse transcription loop-mediated isothermal amplification (RT-LAMP) and (d) real-time RT-LAMP [104]. Nucleic amplification tests (NAAT) are usually preferred as in the case of MERS-CoV diagnosis as it has the highest sensitivity at the earliest time point in the acute phase of infection [102]. Chinese health authorities have recently posted the full genome of 2019-nCoV in the GenBank and in GISAID portal to facilitate in the detection of the virus [11]. Several laboratory assays have been developed to detect the novel coronavirus in Wuhan, as highlighted in WHO’s interim guidance on nCoV laboratory testing of suspected cases. These include protocols from other countries such as Thailand, Japan and China [105].

The first validated diagnostic test was designed in Germany. Corman et al. had initially designed a candidate diagnostic RT-PCR assay based on the SARS or SARS-related coronavirus as it was suggested that circulating virus was SARS-like. Upon the release of the sequence, assays were selected based on the match against 2019-nCoV upon inspection of the sequence alignment. Two assays were used for the RNA dependent RNA polymerase (RdRP) gene and E gene where E gene assay acts as the first-line screening tool and RdRp gene assay as the confirmatory testing. All assays were highly sensitive and specific in that they did not cross-react with other coronavirus and also human clinical samples that contained respiratory viruses [11].

The Hong Kong University used two monoplex assays which were reactive with coronaviruses under the subgenus Sarbecovirus (consisting of 2019-nCoV, SARS-CoV and SARS-like coronavirus). Viral RNA extracted from SARS-CoV can be used as the positive control for the suggested protocol assuming that SARS has been eradicated. It is proposed that the N gene RT-PCR can be used as a screening assay while the Orf1b assay acts as a confirmatory test. However, this protocol has only been evaluated with a panel of controls with the only positive control SARS-CoV RNA. Synthetic oligonucleotide positive control or 2019-nCoV have yet to be tested [106].

The US CDC shared the protocol on the real time RT-PCR assay for the detection of the 2019-nCoV with the primers and probes designed for the universal detection of SARS-like coronavirus and the specific detection of 2019-nCoV. However, the protocol has not been validated on other platforms or chemistries apart from the protocol described. There are some limitations for the assay. Analysts engaged have to be trained and familiar with the testing procedure and result interpretation. False negative results may occur due to insufficient organisms in the specimen resulting from improper collection, transportation or handling. Also, RNA viruses may show substantial genetic variability. This could result in mismatch between the primer and probes with the target sequence which can diminish the assay performance or result in false negative results [107]. Point-of-care test kit can potentially minimize these limitations, which should be highly prioritized for research and development in the next few months.

#### 4.1.2. Serological Testing

Serological testing such as ELISA, IIFT and neutralization tests are effective in determining the extent of infection, including estimating asymptomatic and attack rate. Compared to the detection of viral genome through molecular methods, serological testing detects antibodies and antigens. There would be a lag period as antibodies specifically targeting the virus would normally appear between 14 and 28 days after the illness onset [108]. Furthermore, studies suggest that low antibody titers in the second week or delayed antibody production could be associated with mortality with a high viral load. Hence, serological diagnoses are likely used when nucleic amplification tests (NAAT) are not available or accessible [102].

### 4.2. Vaccines

Vaccines can prevent and protect against infection and disease occurrence when exposed to the specific pathogen of interest, especially in vulnerable populations who are more prone to severe outcomes. In the context of the current 2019-nCoV outbreak, vaccines will help control and reduce disease transmission by creating herd immunity in addition to protecting healthy individuals from infection. This decreases the effective R0 value of the disease. Nonetheless, there are social, clinical and economic hurdles for vaccine and vaccination programmes, including (a) the willingness of the public to undergo vaccination with a novel vaccine, (b) the side effects and severe adverse reactions of vaccination, (c) the potential difference and/or low efficacy of the vaccine in populations different from the clinical trials’ populations and (d) the accessibility of the vaccines to a given population (including the cost and availability of the vaccine).

Vaccines against the 2019-nCoV are currently in development and none are in testing (at the time of writing). On 23 January 2020, the Coalition for Epidemic Preparedness Innovations (CEPI) announced that they will fund vaccine development programmes with Inovio, The University of Queensland and Moderna, Inc respectively, with the aim to test the experimental vaccines clinically in 16 weeks (By June 2020). The vaccine candidates will be developed by the DNA, recombinant and mRNA vaccine platforms from these organizations [109].

Based on the most recent MERS-CoV outbreak, there are already a number of vaccine candidates being developed but most are still in the preclinical testing stage. The vaccines in development include viral vector-based vaccine, DNA vaccine, subunit vaccine, virus-like particles (VLPs)-based vaccine, inactivated whole-virus (IWV) vaccine and live attenuated vaccine. The latest findings for these vaccines arebased on the review by Yong et al. (2019) in August 2019 [110]. As of the date of reporting, there is only one published clinical study on the MERS-CoV vaccine by GeneOne Life Science & Inovio Pharmaceuticals [47]. There was one SARS vaccine trial conducted by the US National Institute of Allergy and Infectious Diseases. Both Phase I clinical trials reported positive results, but only one has announced plans to proceed to Phase 2 trial [111].

Due to the close genetic relatedness of SARS-CoV (79%) with 2019-nCoV [112], there may be potential cross-protective effect of using a safe SARS-CoV vaccine while awaiting the 2019-nCoV vaccine. However, this would require small scale phase-by-phase implementation and close monitoring of vaccinees before any large scale implementation.

### 4.3. Therapeutics

Apart from the timely diagnosis of cases, the achievement of favorable clinical outcomes depends on the timely treatment administered. ACE2 has been reported to be the same cell entry receptor used by 2019-nCoV to infect humans as SARS-CoV [113]. Hence, clinical similarity between the two viruses is expected, particularly in severe cases. In addition, most of those who have died from MERS-CoV, SARS-CoV and 2019-nCoV were advance in age and had underlying health conditions such as hypertension, diabetes or cardiovascular disease that compromised their immune systems [114]. Coronaviruses have error-prone RNA-dependent RNA polymerases (RdRP), which result in frequent mutations and recombination events. This results in quasispecies diversity that is closely associated with adaptive evolution and the capacity to enhance viral-cell entry to cause disease over time in a specific population at-risk [115]. Since ACE2 is abundantly present in humans in the epithelia of the lung and small intestine, coronaviruses are likely to infect the upper respiratory and gastrointestinal tract and this may influence the type of therapeutics against 2019-nCoV, similarly to SAR-CoV.

However, in the years following two major coronavirus outbreaks SARS-CoV in 2003 and MERS-CoV in 2012, there remains no consensus on the optimal therapy for either disease [116,117]. Well-designed clinical trials that provide the gold standard for assessing the therapeutic measures are scarce. No coronavirus protease inhibitors have successfully completed a preclinical development program despite large efforts exploring SARS-CoV inhibitors. The bulk of potential therapeutic strategies remain in the experimental phase, with only a handful crossing the in vitro hurdle. Stronger efforts are required in the research for treatment options for major coronaviruses given their pandemic potential. Effective treatment options are essential to maximize the restoration of affected populations to good health following infections. Clinical trials have commenced in China to identify effective treatments for 2019-nCoV based on the treatment evidence from SARS and MERS. There is currently no effective specific antiviral with high-level evidence; any specific antiviral therapy should be provided in the context of a clinical study/trial. Few treatments have shown real curative action against SARS and MERS and the literature generally describes isolated cases or small case series.

Many interferons from the three classes have been tested for their antiviral activities against SARS-CoV both in vitro and in animal models. Interferon β has consistently been shown to be the most active, followed by interferon α. The use of corticosteroids with interferon alfacon-1 (synthetic interferon α) appeared to have improved oxygenation and faster resolution of chest radiograph abnormalities in observational studies with untreated controls. Interferon has been used in multiple observational studies to treat SARS-CoV and MERS-CoV patients [116,117]. Interferons, with or without ribavirin, and lopinavir/ritonavir are most likely to be beneficial and are being trialed in China for 2019-nCoV. This drug treatment appears to be the most advanced. Timing of treatment is likely an important factor in effectiveness. A combination of ribavirin and lopinavir/ritonavir was used as a post-exposure prophylaxis in health care workers and may have reduced the risk of infection. Ribavirin alone is unlikely to have substantial antiviral activities at clinically used dosages. Hence, ribavirin with or without corticosteroids and with lopinavir and ritonavir are among the combinations employed. This was the most common agent reported in the available literature. Its efficacy has been assessed in observational studies, retrospective case series, retrospective cohort study, a prospective observational study, a prospective cohort study and randomized controlled trial ranging from seven to 229 participants [117]. Lopinavir/ritonavir (Kaletra) was the earliest protease inhibitor combination introduced for the treatment of SARS-CoV. Its efficacy was documented in several studies, causing notably lower incidence of adverse outcomes than with ribavirin alone. Combined usage with ribavirin was also associated with lower incidence of acute respiratory distress syndrome, nosocomial infection and death, amongst other favorable outcomes. Recent in vitro studies have shown another HIV protease inhibitor, nelfinavir, to have antiviral capacity against SARS-CoV, although it has yet to show favorable outcomes in animal studies [118]. Remdesivir (Gilead Sciences, GS-5734) nucleoside analogue in vitro and in vivo data support GS-5734 development as a potential pan-coronavirus antiviral based on results against several coronaviruses (CoVs), including highly pathogenic CoVs and potentially emergent BatCoVs. The use of remdesivir may be a good candidate as an investigational treatment.

Improved mortality following receipt of convalescent plasma in various doses was consistently reported in several observational studies involving cases with severe acute respiratory infections (SARIs) of viral etiology. A significant reduction in the pooled odds of mortality following treatment of 0.25 compared to placebo or no therapy was observed [119]. Studies were however at moderate to high risk of bias given their small sample sizes, allocation of treatment based on the physician’s discretion, and the availability of plasma. Factors like concomitant treatment may have also confounded the results. Associations between convalescent plasma and hospital length of stay, viral antibody levels, and viral load respectively were similarly inconsistent across available literature. Convalescent plasma, while promising, is likely not yet feasible, given the limited pool of potential donors and issues of scalability. Monoclonal antibody treatment is progressing. SARS-CoV enters host cells through the binding of their spike (S) protein to angiotensin converting enzyme 2 (ACE2) and CD209L [118]. Human monoclonal antibodies to the S protein have been shown to significantly reduce the severity of lung pathology in non-human primates following MERS-CoV infection [120]. Such neutralizing antibodies can be elicited by active or passive immunization using vaccines or convalescent plasma respectively. While such neutralizing antibodies can theoretically be harvested from individuals immunized with vaccines, there is uncertainty over the achievement of therapeutic levels of antibodies.

Other therapeutic agents have also been reported. A known antimalarial agent, chloroquine, elicits antiviral effects against multiple viruses including HIV type 1, hepatitis B and HCoV-229E. Chloroquine is also immunomodulatory, capable of suppressing the production and release of factors which mediate the inflammatory complications of viral diseases (tumor necrosis factor and interleukin 6) [121]. It is postulated that chloroquine works by altering ACE2 glycosylation and endosomal pH. Its anti-inflammatory properties may be beneficial for the treatment of SARS. Niclosamide as a known drug used in antihelminthic treatment. The efficacy of niclosamide as an inhibitor of virus replication was proven in several assays. In both immunoblot analysis and immunofluorescence assays, niclosamide treatment was observed to completely inhibit viral antigen synthesis. Reduction of virus yield in infected cells was dose dependent. Niclosamide likely does not interfere in the early stages of virus attachment and entry into cells, nor does it function as a protease inhibitor. Mechanisms of niclosamide activity warrant further investigation [122]. Glycyrrhizin also reportedly inhibits virus adsorption and penetration in the early steps of virus replication. Glycyrrhizin was a significantly potent inhibitor with a low selectivity index when tested against several pathogenic flaviviruses. While preliminary results suggest production of nitrous oxide (which inhibits virus replication) through induction of nitrous oxide synthase, the mechanism of Glycyrrhizin against SARS-CoV remains unclear. The compound also has relatively lower toxicity compared to protease inhibitors like ribavirin [123]. Inhibitory activity was also detected in baicalin [124], extracted from another herb used in the treatment of SARS in China and Hong Kong. Findings on these compounds are limited to in vitro studies [121,122,123,124].

Due to the rapidly evolving situation of the 2019-nCoV, there will be potential limitations to the systematic review. The systematic review is likely to have publication bias as some developments have yet to be reported while for other developments there is no intention to report publicly (or in scientific platforms) due to confidentiality concerns. However, this may be limited to only a few developments for review as publicity does help in branding to some extent for the company and/or the funder. Furthermore, due to the rapid need to share the status of these developments, there may be reporting bias in some details provided by authors of the scientific articles or commentary articles in traditional media. Lastly, while it is not viable for any form of quality assessment and meta-analysis of the selected articles due to the limited data provided and the heterogeneous style of reporting by different articles, this paper has provided a comprehensive overview of the potential developments of these pharmaceutical interventions during the early phase of the outbreak. This systematic review would be useful for cross-check when the quality assessment and meta-analysis of these developments are performed as a follow-up study. 

## 5. Conclusions

Rapid diagnostics, vaccines and therapeutics are key pharmaceutical interventions to limit transmission of respiratory infectious diseases. Many potential developments on these pharmaceutical interventions for 2019-nCoV are ongoing in the containment phase of this outbreak, potentially due to better pandemic preparedness than before. However, lessons from MERS-CoV and SARS-CoV have shown that the journeys for these developments can still be challenging moving ahead.

## Figures and Tables

**Figure 1 jcm-09-00623-f001:**
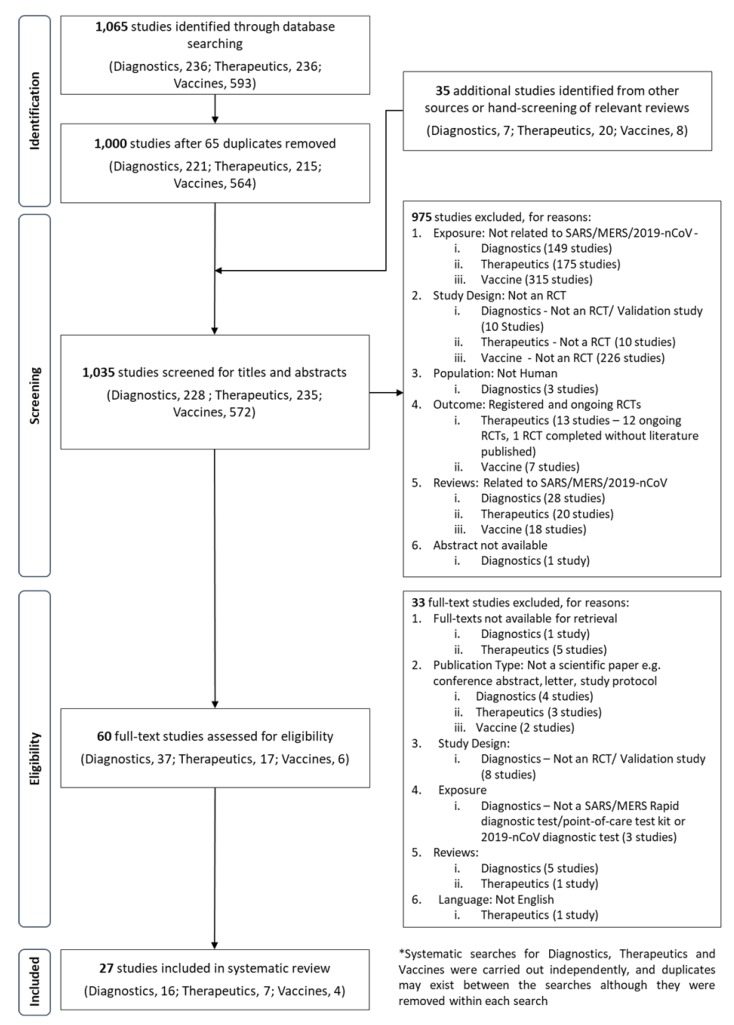
A PRISMA flow diagram of the search strategy for diagnostics, vaccine and therapeutics of 2019-nCoV, MERS-CoV and SARS-CoV.

**Table 1 jcm-09-00623-t001:** Systematic search outcomes for diagnostic evaluation of 2019-nCoV, SARS-CoV and MERS-CoV.

S/N	Year	Author (Country)	Test	Population/Samples	Finding	References
2019-nCoV						
1	2020	Corman et al. (Germany)	rRT-PCR; First-line screening tool: E gene assay; Confirmatory testing: RdRp gene assay.	Respiratory samples were obtained during 2019 from patients hospitalised at Charité medical Centre. Additional samples were selected from biobanks at the Rijksinstituut voor Volksgezondheid en Milieu), Bilthoven, at Erasmus University Medical Center, Rotterdam, at Public Health England, London, and at the University of Hong Kong.	Preliminary finding with SARS-CoV strain Frankfurt-1 virions grown on Vero cells, E gene and RdRp gene assays produced the best result (5.2 and 3.8 copies per reaction at 95% detection probability, respectively) In vitro transcript RNA identical to 2019-nCoV target sequence, 3.9 copies per reaction for the E gene assay and 3.6 copies per reaction for the RdRp assay which were close to the 95% hit rate of 2.9 copies per reaction When tested for cross-reaction with other coronavirus, there was no reactivity with the assays.	[11]
MERS-CoV						
2	2016	Kim et al. (Korea)	Six different commercial MERS-CoV RNA detection kits based on rRT-PCR: (i) PowerChek (Kogene Biotech, Korea); (ii) DiaPlexQ (SolGent, Korea); (iii) Anyplex (Seegene, Korea) Screening: envelope gene (upE) Confirmation: ORF1a (iv) AccuPower (Bioneer, Korea) (v) LightMix (Roche Molecular Diagnostics, Switzerland) (vi) UltraFast kits (Nanobiosys, Korea) detect upE and ORF1a simultaneously	28 nasopharyngeal swabs that were positive for other respiratory viruses were used for specificity and 18 lower respiratory specimens for clinical sensitivity.	All six kits correctly identified 8 of 8 (100%) positive clinical specimens. However, based on the findings from the high inhibition panel, PowerChek and AccuPower were the least sensitive to the presence of PCR inhibition.	[20]
3	2014	Shirato et al. (Japan)	Loopamp RNA Amplification Kit (RT-LAMP; Eiken, Tokyo, Japan)	laboratory isolates MERS-CoV diluted with medium containing pharyngeal swabs obtained from healthy adults	Real-time RT-PCR was able to detect at level as low as 1.6 copies of MERS-CoV RNA while RT-LAMP was also able to detect viral RNA at levels as low as 0.7 copies, showing equivalence with the RT-PCR assay. There was no cross reactivity with other respiratory viruses, for RT-LAMP	[21]
4	2017	Go et al. (Korea)	RT-iiPCR Target upE and ORF1a gene	55 sequential sputum samples collected from 12 patients infected with MERS-CoV were obtained from the Chungnam National University Hospital.	Overall agreement between RT-iiPCR assays and reference RT-qPCR assays were 98.06% (95% CI, 94.43–100%) and 99.03% (95% CI, 95.88–100%) for ORF1a and upE assays, respectively.	[22]
5	2015	Hashemzadeh et al. (Iran)	A onestep rRT-PCR assay, based on specific TaqMan probes	UpE and ORF1b was synthesized due to the difficulty in acquiring patient sample	The sensitivity obtained for upE was fewer than ten copies of RNA template per reaction and for ORF1b was 50 or fewer copies per reaction.	[23]
SARS-CoV						
6	2005	Lau et al. (Hong Kong)	Real time qRT-PCR; Antibody-based capture ELISA	40 SARS patients hospitalized in HK between March–May 2003	Sensitivities for qRT-PCR (80% for fecal samples and 25% for urine samples) were higher than those of the polyclonal (50% and 5%) and monoclonal (35% and 8%) antibody-based nucleocapsid antigen capture enzyme-linked immunosorbent assays.	[24]
7	2003	Lau et al. (Hong Kong)	Enhanced real-time fluorescent PCR	80 suspected or probable SARS cases between 1–3 April 2003	The limit of detection of the enhanced real-time PCR method was 10^2^-fold higher than the standard real-time PCR assay and 10^7^-fold higher than conventional PCR methods In the clinical aspect, the enhanced real-time PCR method was able to detect 6 cases of SARS-CoV positive samples that were not confirmed by any other assay	[25]
8	2004	Jiang et al. (Taiwan)	Quantitative, real-time, nested polymerase chain reaction (PCR)	46 patients with suspected or reported SARS April through May 2003 in Taiwan	The single round PCR yielded a minor amplification signal. Nested PCR produce signal without apparent background. The single round RT-PCR detected 15 of 46 positive cases, while the nest real-time PCR detected 17 of 46 cases.	[26]
9	2004	Wu et al. (Taiwan)	Neutralization test, (ELISA), (IFA), and (ICT)	537 probable SARS patients in Taiwan	With the neutralization test as a reference method, the sensitivity, specificity, positive predictive value, and negative predictive value of the test were ELISA: 98.2%, 98.7%, 98.7%, and 98.4% IFA: 99.1%, 87.8%, 88.1% and 99.1% ICT: 33.6%, 98.2%, 95.7%, and 56.1% RT-PCR: 52.2%, 78.7%, 74.5%, and 58.1%,	[27]
10	2003	He et al. (Singapore)	Western blot assay with N195 protein	274 clinical sera which were collected from patients suffering from probable or suspected SARS, dengue fever, autoimmune diseases	The specificity and sensitivity of this test were 98.3 and 90.9% and 40 of 44 clinical SARS samples were positive.	[28]
11	2003	Poon et al. (Hong Kong)	Real time quantitative RT-PCR modified RNA extraction method 1b region of SARS-CoVCoV	50 NPA samples collected from days 1–3 of disease onset from SARS patients in whom SARS CoV infections was subsequently serologically confirmed	From the 50 NPA specimens collected during the first 3 days of illness, the first-generation RT-PCR assay identified 22% positive sample, modification in the RNA extraction method identified 44% positive samples and the combination of the modified RNA extraction method and real-time quantitative PCR technology, identified 80% positive sample.	[29]
12	2004	Guan et al. (Hong Kong)	Specific ELISA Genelabs Diagnostics Pte Ltd. utilizing two recombinant proteins (Gst-N and Gst-U274)	227 clinical serum specimens collected from SARS patients in Hong Kong between 18 March–24 May 2003 and 385 samples from healthy donors.	For the ELISA, the overall sensitivity was 71.8% and specificity was 99.5% For the immunocromatographic test, overall rate of detection of SARS-associated specimens by the rapid test was 70.5% and its specificity was 97.7%.	[30]
13	2003	Hui et al. (Hong Kong)	RT-PCR	Clinical samples, NPAs (n = 131) and stool specimens (n = 5), provided by the Department of Microbiology, The University of Hong Kong.	PCR amplifying the N gene gave an average of a 26.0% (6.3 to 60.0%) stronger intensity signal than that for the 1b gene additional sensitive molecular marker for the diagnosis of the SARS coronavirus	[31]
14	2004	Mahony et al. (Canada)	Seven RT-PCR assays include (i) a nested assay with BNI outer and inner primers with polB gene; (ii) a two-step, non-nested assay with the BNI outer primers with polB gene; (iii) a two-step, non-nested RT-PCR assay with Cor-p-F2 and Cor-p-R1 with polB gene; (iv) a one-step RT-PCR, with BNI outer primers and polB gene with SYBR Green detection; (v) a two-step assay amplifying and nucleocapsid gene with SYBR Green detection; (vi) a one-step assay with the same nucleocapsid primers and gene with TaqMan probe and (vii) a commercial RealArt HPA CoV RT-PCR assay (Artus).	68 specimens, including 17 respiratory tract specimens (nasopharyngeal or throat swabs), 29 urine samples, and 22 stool samples, were collected between March–April of 2003 from hospitalized patients with a probable or suspected diagnosis of SARS at Sunnybrook and Women’s College Health Sciences Centre during the Toronto outbreak of SARS.	There is no significant difference in the sensitivity and specificity for the 7 assays Assay 1: 94%; 100% Assay 2: 100%; 94% Assay 3: 94%; 100% Assay 4: 100%; 96% Assay 5: 94%; 100% Assay 6: 83%; 100% Assay 7: 94%; 100%	[32]
15.	2004	Liu et al. (Taiwan)	Indirect IFA	Throat wash samples from 17 confirmed SARS adult patients, and 10 healthy controls admitted to the emergency department of the National Taiwan University Hospital between 16 April–1 May 2003.	SARS-CoV was detected in 11 of 17 (65%) samples from SARS patients from day 2 to day 9 of the illness but in none of the 10 samples from healthy controls	[33]
16.	2004	Lin et al. (Taiwan)	SARS-CoV real-time PCR assay with a TaqMan minor groove binder probe developed by Applied Biosystems (Foster City, CA, USA).	228 samples (137 sputum, 53 throat swabs or throat wash, 17 NPS, 19 stool specimen, 2 pleural fluid, 2 urine and 1 serum sample) from 151 patients with atypical pneumonia or symptoms mimicking SARS between 30 April–26 June 26 were recruited, In total, from 151 patients were tested.	• The real time PCR has a threshold sensitivity of 10 genome equivalents per reaction and it has a good reproducibility with the inter-assay coefficients of variation of 1.73 to 2.72%. • 13 specimens from 6 patients were positive with viral load range from 362 to 36,240,000 genome equivalents/mL. The real-time RT-PCR reaction was more sensitive than the nested PCR reaction, as the detection limit for the nested PCR reaction was about 10^3^ genome equivalents in the standard cDNA control.	[34]

Real-time reverse-transcription PCR (rRT-PCR); RNA-dependent RNA polymerase (RdRp); open reading frame 1a (ORF1a); Loop-mediated isothermal amplification (LAMP); enzyme-linked immunosorbent assay (ELISA); immunofluorescent assay (IFA); immunochromatographic test (ICT); nasopharyngeal aspirate (NPA).

**Table 2 jcm-09-00623-t002:** Potential commercial rapid diagnostic kits for 2019-nCoV.

S/N	Type	Organisation	Date	Test	Sensitivity	Specificity	Availability	Turn Around	Costs	Reference
1	RT-PCR	Genesig (U.K.)	Jan 2020	RT-PCR Kit MasterMix and q16 reaction tubes included (for genesig^®^ q16); PCR MasterMix Kit (for other instruments)	Sensitive to < 100 copies of target Broad dynamic detection range (>6 logs)	Stated to be high but with no accompanying statistics	Sold for research use only and are not licensed for diagnostic procedures	(no info)	For use with genesig q16 tubes: £5.44/test; For use with others: £4.62/test	[35]
2	RT-PCR	Bioperfectus Technologies (China)	14 Jan 2020	RT-PCR test kit	Not stated	Not stated	Available as scientific research product—does not require registration	(no info)	(no info)	[36]
3	RT-PCR	Co-Diagnostics (U.S.A.)	23 Jan 2020	Commercial Kit RT-PCR kit	Stated to be high but with no accompanying statistics.	Claims with lower false positive^1^	By Mar 2020	(no info)	(no info)	[37,38]
4	RT-PCR	altona Diagnostics (Germany)	23 Jan 2020	Commercial Kit RT-PCR kit	Not stated	Not stated	(no info)	(no info)	(no info)	[39]
5	RT-PCR	BGI; Pathomics Health (distributor; China) *	23 Jan 2020	Fluorescent RT-PCR kit In vitro RT-PCR combining fluorescent probing ^1^	Not stated	Not stated	Currently used in hospitals and local disease control centres in China. Potentially in Hong Kong, Taiwan, Brunei, Thailand, Nigeria, South Africa too.	No data stated but described as ‘can issue results in a few hours’.	(no info)	[40]
6	Combined RT-PCR and meta- genomics detection	BGI; Pathomics Health (distributor; China) *	23 Jan 2020	2019-nCoV PMseq Kit A metagenomics sequencing kit based on combinatorial Probe Anchor Sythesis. Able to detect both known and novel microorganisms, Enabling monitoring of evolution during transmission ^2^.	Not stated	Not stated	Has been Providing technical support for the scientific clinical prevention and control of the epidemic in Wuhan.	Faster than Fluoresce nt RT- PCR kit. 128 Samples with SE50 in 5 h; 128 samples with PE100 in 22 h,	(no info)	[41]
7	Microfluidic	Veredus Laboratories (Singapore)	24 Jan 2020	enVision (enzyme-assisted nanocomplexes for visual identification of nucleic acids) Lab-on-Chip platform integrating PCR and microarray	Stated to be high but with no accompanying statistics.	Stated to be high but with no accompanying statistics.	(no info)	2 h	(no info)	[42,43]

^1^ BGI, pathomics Health. Real-Time Flourescent RT-PCR kit for detecting 2019-nCoV. Published 2020. ^2^ BGI, pathomics Health. BGI RT-PCR Kit and PMseq^®^ Solution for Detecting 2019-nCoV. Published 2020. * Passed emergency approval procedure of the National Medical Products Administration.

**Table 3 jcm-09-00623-t003:** Ongoing vaccine development for 2019-nCoV.

S/N	Company	Estimated Timeline	Technology	Stage/Funding	Reference
1	Moderna Therapeutics—US National Institute of Allergy and Infectious Diseases	3 months to early stage (phase 1) clinical trial in US (earliest); much longer for full testing and regulatory approval	Messenger RNA vaccine	Preclinical Awaiting preclinical tests and phase 1 study by NIAID, Funding by CEPI.	[62]
2	Inovio Pharmaceuticals	Human testing in the next few months	INO-4800-DNA based vaccine (DNA synthesized in lab, does not require actual virus sample)	Preclinical Funding by Coalition for Epidemic Preparedness Innovations (CEPI), up to $9 million	[62,63]
3	Novavax	3 months	Nanoparticle vaccine	Preclinical	[62]
4	University of Queensland	6 months	Rapid Response Technology, ‘Molecular clamp’ vaccine platform (gene added to viral proteins, misleads body to generate antibodies)	Preclinical Funding by Coalition for Epidemic Preparedness Innovations (CEPI)	[64,65]
5	Vir Biotechnology	Not available	Anti-coronavirus monoclonal antibodies. Additionally, using “whole-genome CRISPR-based screening capabilities to identify the host receptor for Wuhan coronavirus”	Preclinical	[66,67]
6	Chinese Centre for Disease Control and Prevention (CDC)	At least 1 month for development, 2–3 years before availability for use	Not available Inactivated virus vaccine (postulated, not verified)	Preclinical; virus successfully isolated, currently selecting strain	[68,69,70,71]
7	Shanghai East Hospital (Tongji University)—Stermirna Therapeutics	<40 days for manufacture of vaccine samples	mRNA technology	Preclinical	[72]
8	Johnson & Johnson	1 year to market	Adenovirus—vectored technology used for Ebola vaccine (and Zika and HIV vaccine candidates)	Preclinical	[73,74]
9	University of Hong Kong	Months for animal testing, At least 1 year for clinical trials on humans	Modified nasal spray influenza vaccine (with surface antigen of coronavirus) prevents both influenza and corona virus	Preclinical; vaccine developed	[70]
10	University of Saskatchewan (VIDO-InterVac)	Target for animal testing in 6–8 weeks, human trials in at least a year	Not available	Preclinical	[75]
11	GeoVax—BravoVax	Not available	Modified Vaccina Ankara—Virus Like Particles (MVA-VLP) vaccine platform	Preclinical	[76]
12	Clover Biopharmaceuticals	Not available	Highly purified recombinant 2019-nCoV S protein subunit-trimer vaccine (S-Trimer), produced using Trimer-Tag© technology	Preclinical	[77]
13	CureVac	Not available	mRNA technology	Preclinical	[78]
14	Texas Children’s Hospital Center for Vaccine Development at Baylor College of Medicine	Not available	Not available	Not available	[79]
15	Codagenix	Not available	Not available	Not available	[79]

**Table 4 jcm-09-00623-t004:** Systematic search of completed vaccine trial of SARS-CoV and MERS-CoV.

S/N	Organisation; Candidate	Country of Study; Trial Type; Study Design; Study Details	Population	Outcome (Safety)	Outcome (Efficacy)	Reference
SARS-CoV						
1	Sinovac Biotech Co. Ltd.; Inactivacted SARS-CoV (ISCV)	China (Beijing); Phase I clinical trial; Randomised, double-blind and placebo controlled; 2 doses of 16 SARS-CoV units (SU) or 32 SU ISCV or placebo control vaccine, intramuscular injection of vaccines in deltoid muscle, doses were 28 days apart	36 healthy adults between 21 and 40 years old, free of chronic diseases, immunosuppression SARS-CoV, HCV and HIV; 12 subjects were included in each intervention arm	No severe adverse reaction (grade 3) was reported. All local adverse events were mild and resolved within 47–72 h, while systemic adverse events were reported sporadically from all 3 groups and resolved within 24 h.	Seroconversion reached 100% for both vaccine groups on day 42, persisted at 100% in the group receiving 16 SU but decreased to 91.1% for the group receiving 32 SU on day 56. Geometric mean titres (GMT) of specific SARS-CoV neutralising antibody peaked 2 weeks after the second dose, but started to drop 4 weeks later (values not reported). Seroconversion and GMT of neutralising antibody levels were lower in subjects between 21–30 years old compared to those in the elder group, but without significant differences (35: seroconversion *p* = 0.444; GMT P = 0.528) on days 35 and 42.	[44]
2	National Institutes of Health, National Institute of Allergy and Infectious Diseases, Vaccine Research Center; VRC-SRSDNA015-00-VP	United States (Maryland) Phase I clinical trial; Open-label study; 3 doses of vaccine (4 mg/dose), intramuscular injection into lateral deltoid muscle via the Biojector 2000^®^ Needle-Free Injection Management System™ on study days 0, 38 and 56	10 subjects between 21 and 29 years old (mean age 35.5) with mean BMI of 24.6 (range 19.7 to 33.9) and were Caucasians (90%) or Asian (10%); only 9 subjects completed all 3 doses	No severe adverse reaction (grade 3), but at least 50% subjects reported at least one mild systemic symptom following vaccination.	SARS specific antibody was detected by ELISA in 8 of 10 (80%) of subjects at one or more timepoints. The neutralizing antibody response was detected in all subjects who received 3 doses of vaccine and peaked between week 8 and 12, with 6 subjects remaining positive at week 32. SARS-CoV-specific CD4+ T cell responses were detected in all vaccines between week 2 and 32, and CD8+ T cell responses in ∼20% of individuals by ICS. The peak T cell response occurred between week 8 and 12 and when present, was sustained throughout the 32 week trial.	[45]	
MERS-CoV						
4	GeneOne Life Science and Inovio Pharmaceuticals; GLS-5300	United States (Maryland) Phase I clinical trial; Open-label, single-arm, dose-escalation study; 3 doses of 0·67 mg, 2 mg, or 6 mg GLS-5300; intramuscular 1 mL injection followed immediately by co-localised intramuscular electroporation with CELLECTRA^®^-5P device at week 0, 4 and 12 with follow-up through to 48 weeks after dose 3	75 healthy adults between 18 and 50 years old (mean age 32.2 years), with normal screening electrocardiogram, screening laboratory findings within normal limits or be grade 0–1 findings, and have no history of clinically significant immunosuppressive or autoimmune disease, HIV, hepatitis B or C virus infection; 25 subjects were randomised into each dose group and 65 subjects completed the study, and per protocol analysis was used.	No vaccine-associated serious adverse events. 97% participants reported at least one solicited adverse event, but most solicited symptoms were reported as mild and were self-limiting (19 [76%] with 0·67 mg, 20 [80%] with 2 mg, and 17 [68%] with 6 mg); injection site reactions were the most common adverse event [92%].	Seroconversion measured by S1-ELISA occurred in 86% and 94% participants after 2 and 3 doses, respectively, and was maintained in 79% participants up to study end at week 60. Neutralising antibodies were detected in 50% participants at one or more time points during the study, but only 3% maintained neutralisation activity to end of study. T-cell responses were detected in 71% and 76% participants after 2 and 3 doses, respectively. There were no differences in immune responses between dose groups after 6 weeks and vaccine-induced humoral and cellular responses were respectively detected in 77% and 64% participants at week 60.	[47]

**Table 5 jcm-09-00623-t005:** Potential & ongoing therapeutics trials of 2019-nCoV infection.

S/N	Company	Treatment	Stage	Reference
1	AbbVie	Lopinavir-ritonavir	Approved. Used in clinical settings. The Jin Yintan Hospital in Wuhan, China, launched a randomised, open-label, blank-controlled trial for the efficacy and safety of lopinavir-ritonavir and interferon-alpha 2b in hospitalisation of 80 patients with novel coronavirus infection. Lopinavir-ritonavir tablets (each containing 200 mg of lopinavir and 50 mg of ritonavir), twice a day, 2 tablets at a time; interferon-α2b. Assessment of effectiveness of treatment based on clinical improvement time of 28 days after randomisation.	[80]
2	Sanofi-Aventis	Teicoplanin (Targocid)	Approved. Used in clinical setting.	
3	Gilead Science	Remdesivir	Gilead is in active discussions with researchers and clinicians in the United States and China regarding the ongoing Wuhan coronavirus outbreak and the potential use of remdesivir as an investigational treatment.	[81]
4	Vir	Monoclonal antibodies	Vir is working to rapidly determine whether its previously identified anti-coronavirus monoclonal antibodies (mAbs) bind and neutralize 2019-nCoV.	[67]
5	Regeneron	Monoclonal antibodies	Regeneron Pharmaceuticals has developed monoclonal antibodies to treat MERS that are now being tested in early human studies. A company spokesperson said that researchers have begun to identify similar antibodies that might work against 2019-nCoV. With Ebola, it took Regeneron six months to develop candidate treatments and test them in animal models.	[82]
6	Ascletis	Ritonavir + ASC09 combo	Applied to include in national emergency channel on 25 January 2020. Not yet approved by regulators.	[83]
7	Biocryst Pharmaceuticals	Galidesivir	Biocryst is evaluating Galidesivir to determine if it could potentially target the coronavirus. Galidesivir is currently in a phase 1 clinical study.	[84]
8	Purdue University	Molecules that inhibit 2 coronavirus enzymes	Molecules developed by the university scientists inhibit two coronavirus enzymes and prevent its replication. The discovered drug targets are said to be more than 95% similar to enzyme targets found on the SARS virus. Researchers note that identified drugs may not be available to address the ongoing outbreak but they hope to make it accessible for future outbreaks.	[85]
9	The First Affiliated Hospital of Guangzhou Medical University	“xue bi jing” (TCM) -ChiCTR2000029381	Approved. Recruitment of subjects has not started.	[86]
10	Chongqing Public Health Medical Center	Adjunctive steroids has a trial-ChiCTR2000029386	Approved. Recruitment of subjects has not started.	[87]
11	Ruijin Hospital	Umefinovir (arbidol)- NCT04260594	An antiviral treatment for influenza infection. Preliminary test in the in vitro cell showed an effective inhibition of coronavirus and a significant inhibition to the cytopathic effect.	[88]
12	Shanghai Public Health Clinical Center	Darunavir-NCT04252274	An antiviral treatment for HIV. Study showed that it can significantly inhibit the replication of the new coronavirus.	[88]
13	Harbin Pharmaceutical Group Sanjing Pharmaceutical Holding Co., Ltd.	Oral liquid traditional Chinese medicine, Shuanghuanglian -ChiCTR2000029605	Preliminary testing identified that it can inhibit the new coronavirus. It was previously identified to have an antiviral effect for influenza virus, SARS and MERS.	[88,89,90]
14	The Fifth Affiliated Hospital of Sun Yat-Sen University	Chloroquine phosphate -ChiCTR2000029609	Approved. Recruitment of subjects has not started.	[88,91]
15	Shanghai Public Health Clinical Center	Hydroxychloroquine	Recruitment in process; Interventional subjects will receive hydroxychloroquine 400 mg per day for 5 days, also take conventional treatments.	[53]
16	Tongji Hospital	Abidol Hydrochloride combined with Interferon atomization	Recruitment of subjects has not started; Interventional subjects will receive standard symptomatic support therapy (SMT) plus abidol hydrochloride (0.2 g, 3 times a day) or Abidol Hydrochloride combined with Interferon (PegIFN-α-2b) atomization (45 ug)	[55]
17	Tongji Hospital	Drug: Abidol hydrochloride, Oseltamivir, Lopinavir/ritonavir	Recruitment of subjects has not started; Interventional subjects will receive either Abidol hydrochloride 0.2 g once, 3 times a day, 2 weeks or Oseltamivir 75 mg once, twice a day, 2 weeks or Lopinavir/ritonavir 500 mg once, twice a day, 2 weeks	[54]
18	Guangzhou 8th People’s Hospital	Lopinavir Plus Ritonavir and Arbidol	Recruitment in process; Interventional subjects will receive either standard treatment plus a regimen of lopinavir (200 mg) and ritonavir (50 mg) (oral, q12h, every time 2 tablets of each, taking for 7–14 days) or Standard treatment plus a regimen of arbidol (100 mg) (oral, tid, 200 mg each time, taking for 7–14 days).	[56]
19	China-Japan Friendship Hospital	Remdesivir	Recruitment of subjects has not started	[57]
20	Shanghai Public Health Clinical Center	Darunavir & Cobicistat	Recruitment of subjects has not started; Interventional subjects will receive darunavir and cobicistat one tablet per day for 5 days plus conventional treatments.	[58]
21	Beijing 302 Hospital	Mesenchymal Stem Cell (MSC)	Recruitment in process; Interventional subjects will receive conventional treatment plus 3 times of MSCs (0.5–1.0 × 10E6 MSCs/kg body weight intravenously at Day 0, Day 3, Day 6).	[59]
22	Beijing 302 Hospital	Traditional Chinese Medicine	Recruitment in process; Interventional subjects will receive oxygen therapy, antiviral therapy (alfa interferon via aerosol inhalation, and lopinavir/ritonavir, 400 mg/100 mg, p.o, bid) for 14 days plus Traditional Chinese Medicines (TCMs) granules: one bag, p.o, bid, for 14 days.	[60]
23	Medical ICU, Peking Union Medical College Hospital	Methylprednisolone	Recruitment in process; Interventional subjects will receive standard care plus methylprednisolone therapy (40 mg q12h for 5 days)	[61]

**Table 6 jcm-09-00623-t006:** Completed Randomized Controlled Trials for SARS-CoV & MERS-CoV Therapeutics.

S/N	Date	Country	No. of Participants	Treatment	Outcome	Reference
SARS-CoV						
1	Recruitment: 24 March–28 April 2003	Hong Kong	152/152	Historical control: 4 g oral loading dose followed by 1.2 g every 8 h, or 8 mg/kg intravenously every 8 h if the patient could not tolerate oral treatment) with a reducing regimen of corticosteroid for 21 days Treatment group: combination of lopinavir (400 mg)/ritonavir (100 mg) orally every 12 h for 14 days Both groups given ribavirin & corticosteroid according to the same protocol	Primary outcome: composite adverse outcome at 21 days, severe hypoxaemia to fraction of inspired oxygen or death The 21 day adverse outcome rate was therefore 28.8% for the historical controls and 2.4% for the treatment group, giving an effect size of 26.4% (95% confidence interval 16.8 to 36.0, *p* = 0.001) for lopinavir/ritonavir treatment. Apparent reduction in also viral load, steroid dose, incidence of nocosomial infections in treatment group compared to control group.	[98]
2	Recruitment: 24 April–30 June 2003	Taiwan	4/4	CM A: Composition of 13 herbs CM B: Popular health care product in Taiwan	Death in patient 1 (placebo control), recovery in remaining 3 patients	[93]
3	Recruitment: 10 April–31 May 2003	China	115/123	Western medicine: oxygen supplementation, hemofiltration, ribavirin, antibacterials (azithromycin, cefuroxime, metronidazole), and immunoregulation with thymosin injection. Combined treatment: Herba houttuyniae with western medicine, and when necessary, TCM treatment like heat clearing and detoxifying, qi supplementing, blood regulating prescription.	Patients with early symptoms experienced a longer hospital stay (*p* = 0.028), a non-statistically significant mortality rate decrease (combined treatment: 9.6% versus WM: 11.1%), and a significant improvement of arthralgia and myalgia (*p* < 0.05) when on combined treatment compared with a strictly WM treatment. Combined treatment also improved arterial oxyhemoglobin saturation significantly at day 22 (*p* < 0.05) compared to WM.	[96]
4	20 April 2003–30 May 2003	Hong Kong	16/17	9 (Early hydrocortisone treatment) & 7 (Saline placebo)	Median time for SARS-CoV RNA to become undetectable In plasma was 12 days vs. 8 days in the hydrocortisone and placebo groups. Corticosteroid treatment early in the treatment was associated with higher subsequent plasma viral load compared to placebo (AUC; Mann–Whitney, *p* = 0.023), delaying viral clearance	[95]
5	28 January–28 February 2003	China	190/190	40 (A; Ribavirin, cefoperazone/sulbactam); 30 (B; fluoroquinolone, azithromycin, recombinant interferon alpha and restricted steroid use); 60 (C; quinolone, azithromycin, some given recombinant interferon alpha, steroid use when symptoms worsen); 60 (D; levofloxacin, azithromycin, of which 45 were given recombinant interferon alpha)	Treatment D presented with the most favourable outcome. The shortest mean time to discharge, 20.7 (SD, 4.6) days, was observed in treatment group D, compared to 24.8, 24.8 and 22.4 days for treatment groups A, B and C respectively. None of the patients on treatment D needed mechanical ventilation, all recovered and were discharged from the hospital compared to 2, 2, and 7 deaths from patients on treatments A, B and C respectively	[97]
6	Not reported	China	49/52	29 (Control: Ribavirin, Levofloxacin, Thyopentin, Azithromycin, methylpredisolone) 20 (Treatment: Control group treatment + TCM Recipes)	All patients recovered. Significantly shorted time from the disease onset to the symptom improvement in treatment (5.10 ± 2.83 days) compared to control group (7.62 ± 2.27 days) (*p* < 0.05) No significant difference in blood routine improvement, pulmonary chest shadow in chest film improvement and corticosteroid usgae between the 2 groups. However, particularly in the respect of improving clinical symptoms, elevating quality of life, promoting immune function recovery, promoting absorption of pulmonary inflammation, reducing the dosage of cortisteroid and shortening the therapeutic course, treatment with integrative chinese and western medicine treatment had obvious superiority compared with using control treatment alone.	[94]
MERS-CoV						
7	2 June 2016–4 January 2017	United States	38/43	Each group comprises 6 cohorts (Escalating doses 1 mg/kg, 2.5 mg/kg, 5 mg/kg, 10 mg/kg, 20 mg/kg & 50 mg/kg in SAB-001 treatment group)	A total of 97 adverse events (AEs) were reported, with a mean of 2.3 AEs per participant in the SAB-301 group and a mean of 3.3 AEs per participant in the placebo group. No serious adverse event related to the study intervention was observed. Single dose pharmacokinetics demonstrated relatively linear and dose-proportional increases in maximal concentration and area-under-the-concentration-time curve. Microneutralization titres also correlated to the SAB-301 levels in serum. Single infusions of SAB-301 up to 50 mg/kg appear to be safe and well-tolerated in healthy participants.	[46]

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
