# Peer review of "Potential Rapid Diagnostics, Vaccine and Therapeutics for 2019 Novel Coronavirus (2019-nCoV): A Systematic Review"

_jcm, 2020, doi:10.3390/jcm9030623_

Round 1

Reviewer 1 Report

This paper presents compiles information regarding rapid diagnostics, vaccines, and other therapeutics relevant to the current novel coronavirus (COVID-19) epidemic and appears to be very comprehensive. Overall, the English is very good and the manuscript is understandable as is. However, there are many minor grammatical errors and repetitive descriptors (too many adjectives used to describe something) throughout the manuscript. Only some of these issues are pointed out in this review.

Major comments:

Differences in clinical features between COVID-19, SARS, and MERS are not addressed. Such information impacts reader understanding of the utility of SARS and MERS diagnostics, vaccines, and therapeutics to be applied to COVID-19 (particularly with regard to therapeutics, where the authors did not return any results on COVID-19 trials). I highly recommend that the authors address this point.

Minor comments, including some suggested grammatical changes:

Abstract: MERS-CoV, SARS-CoV, PRISMA, and RT-PCR are not defined

Figure 1: figures should be stand-alone. Please define SARS, MERS, PRISMA, and RCT somewhere.

All tables: please be sure to define all acronyms and initialisms (as a footnote is fine). Date ranges should use en-dashes with no spaces.

Line 15: vaccine should be plural (“vaccines”)

Line 23: the RT-PCR (or did the authors mean “rRT-PCR”?) of the Beijing Genomic Institute is only mentioned here in the abstract (based on a keyword search of the manuscript) and does not appear to be cited in the references. At the very least is should be mentioned somewhere in the main text if it is mentioned in the abstract.

Line 32: the statement “This is one key characteristic that may be similar in 2019-nCoV” is unsubstantiated. Please provide more information about why this statement was made.

Line 47: SARS-CoV-2 is inappropriately referred to as “Wuhan virus”

Line 80: number/date ranges starting with the word “between” should use “and” rather than “to”

Lines 96–97: vaccines should be plural

Lines 97–98: “latest” is used twice in the same sentence

Line 110: “Years following the major” --> “In the years following the two recent major”

Line 111: RT-PCR not previously defined in the main text. Please define here. Same for RT-LAMP and RT-iiPCR in the next paragraph.

Line 124: “MERS-Cov” --> “MERS-CoV”

Line 145: “SARS-Cov” --> “SARS-CoV”

Line 148: Use of the term “BatCoVs” is not defined

Line 152: severe acute respiratory infections should not be capitalized

Line 153: viral etiology should not be capitalized

Line 175: “SARS epidemic – 3 in China” --> “SARS epidemic—3 in China” (rather than an en-dash with spaces, it is correct to use an em-dash with no spaces)

Line 192: verb form “to report” is used for two different actions with different tenses. Some developments “have yet to be reported” while for other developments “there is no intention to report” is grammatically correct.

Line 193: concerns should be plural

Line 207: “already ongoing rapidly” --> too many descriptors; “ongoing” would suffice

Line 208: “as before” --> “than before”

Reviewer 2 Report

Very interesting topic. I am sure the need for diagnostics/therapy/vaccine/etc. for the novel coronavirus (COVID-19) is huge in this pandemic status. However, it has only been a month since the first cluster of outbreak in Wuhan, China, has been reported, I do not feel this is the appropriate time to review for diagnostics/therapy/vaccine. I can see from your article that the competition for these new methods around the globe is fierce, but it is a bit too early to review them at this time point. For example, you list a number of possible compounds that may act as anti-virals against COVID-19, but there is no effective drugs that can be used in the clinical settings at the moment, giving little clinical/public health impact to the readers. The same for vaccinations.

Nonetheless, by the time the research on diagnostics/therapy/vaccine has progressed, possibly a few months later, your review will be precious in a way that the readers can systematically review the effectiveness, drawbacks, or any other important characteristicss of diagnostics/therapy/vaccine for the new respiratory virus. 
